# Graphene mechanical pixels for Interferometric Modulator Displays

Santiago J. Cartamil-Bueno[1], Dejan Davidovikj [2], Alba Centeno[3], Amaia Zurutuza[3], Herre S.J. van der Zant[2], Peter G. Steeneken [2] & Samer Houri[2,4]

Electro-optic modulators based on micro-electromechanical systems have found success as elements for optical projectors, for simplified optical spectrometers, and as reflective-type screens that make use of light interference (Interferometric Modulator Display technology). The latter concept offers an exciting avenue for graphene nanomechanical structures to replace classical micro-electromechanical devices and bring about enhancement in performance, especially switching speed and voltage. In this work we study the optical response of electrically actuated graphene drumheads by means of spectrometric and stroboscopic experiments. The color reproducibility and speed of these membranes in producing the desired electro-optic modulation makes them suitable as pixels for high refresh rate displays. As a proof of concept, we demonstrate a Graphene Interferometric Modulator Display prototype with 5 μm-in-diameter pixels that compose a high resolution image (2500 pixels per inch)—equivalent to a 5″ display of 12K—whose color can be changed at frame rates of at least 400 Hz.

[1] SCALE Nanotech OÜ, Sepapaja 6, Tallin 15551, Estonia. [2] Kavli Institute of Nanoscience, Delft University of Technology, Lorentzweg 1, 2628CJ Delft, The Netherlands. [3] Graphenea SA, 20018 Donostia-San Sebastián, Spain. [4]Present address: NTT Basic Research Laboratories, NTT Corporation, 3-1 Morinosato-Wakamiya, Atsugi, Kanagawa 243-0198, Japan. Correspondence and requests for materials should be addressed to S.J.C-B. (email: cartamil@scalenano.tech) or to S.H. (email: Houri.Samer@lab.ntt.co.jp)

Graphene, the carbon monolayer and two-dimensional allotrope of graphite, has the potential to impact technology with a wide range of applications such as optical modulators for high-speed communications[1–4]. In contrast to modulation devices that rely on plasmonic or electronic effects, micro-electromechanical system (MEMS)-based modulators can have wider tuning ranges albeit at a lower operating frequency. These properties make electro-optic mechanical modulators ideal for reflective-type display technologies as has been demonstrated previously with SiN membranes in Interferometric Modulator Displays (IMODs)[5]. Despite their low-power consumption and performance in bright environments, IMODs suffer from low frame rates and limited color gamut.

Double-layer graphene (DLG) membranes grown by chemical vapor deposition (CVD) can also recreate the interference effect like in IMODs as proven with drumheads displaying Newton's rings[6]. Here, we report on the electro-optical response of CVD DLG mechanical pixels by measuring the change in wavelength-dependent reflectance of a suspended graphene drumhead as a function of electrical gating. We use a spectrometer to measure the wavelength spectrum at different voltages, and find a good agreement with a model based on light interference. Moreover, to verify that gas compression effects do not play an important role, we use a stroboscopic illumination technique to study the electro-optic response of these graphene pixels at frequencies up to 400 Hz. Based on these findings, we demonstrate a continuous full-spectrum reflective-type pixel technology with reduced flicker effect by fabricating a Graphene Interferometric Modulator Display (GIMOD) prototype of 2500 pixels per inch (ppi) equivalent to more than 12 K resolution.

## Results

**Fabrication and setup**. Circular cavities are etched through thermally grown $SiO_2$ layers (300 to 1180 nm in depth depending on the sample) on silicon substrates by means of reactive-ion etching. DLG layers, fabricated by stacking two CVD single-layer graphene (SLG) layers, are transferred onto the patterned substrate using a semi-dry transfer technique. This results in an optical cavity with a movable absorbing membrane made out of CVD DLG (Fig. 1a), and a fixed mirror formed by the underlying silicon surface. The silicon substrate also plays the role of back electrode for electrostatic actuation. We use DLG membranes because of their higher yield[7] and their larger absorption in the visible spectrum than that of SLG.

The colorimetry setup used in this work consists of an optical microscope with Köhler illumination, a 20× apochromatic objective lens, and a halogen lamp as a multi-wavelength light source. Light reflected from the sample is split and guided both towards a calibrated consumer camera and a spectrometer. The spectrometer is configured to collect only the light from a circular area with the same size as the studied drums (see Methods). A series of color filters with wavelengths ranging from 450 nm to 650 nm (bandpass linewidth of 5–10 nm) are mounted on a motorized computer-controlled wheel, and placed in front of the camera imager as shown schematically in Fig. 1b.

Electrostatic actuation of the graphene drums is achieved by using a computer-controlled DC source connected to the suspended graphene electrode, while keeping the silicon substrate grounded. For these measurements a sample with 1180 nm oxide thickness was chosen. The capacitance of the graphene/$SiO_2$/Si structure (area of ~1 cm$^2$) was measured to be 0.6 nF. Figure 1c shows the shift in the spectrum upon the application of a voltage. As the voltage is ramped up (see Methods), the overall color of the drumhead changes, going from orange ($\lambda = 580$ nm) at 0 V to green ($\lambda = 550$ nm at 12.5 V and $\lambda = 510$ nm at 20 V).

**Deflection–voltage curves**. The spectral response of the graphene drums allows us to extract the deflection of the drum, which can be used to relate the electromechanical and optomechanical performance of the pixel. By applying a force equilibrium condition between the electrostatic force, the linear elastic force (stiffness $k_1$), the cubic elastic force (stiffness $k_3$), and the hydrostatic force due to the pressure of trapped gas inside the cavity, we develop a 1-degree-of-freedom graphene membrane electromechanical model where the bending rigidity is neglected (detailed mathematical derivations in Supplementary Notes 2–4,

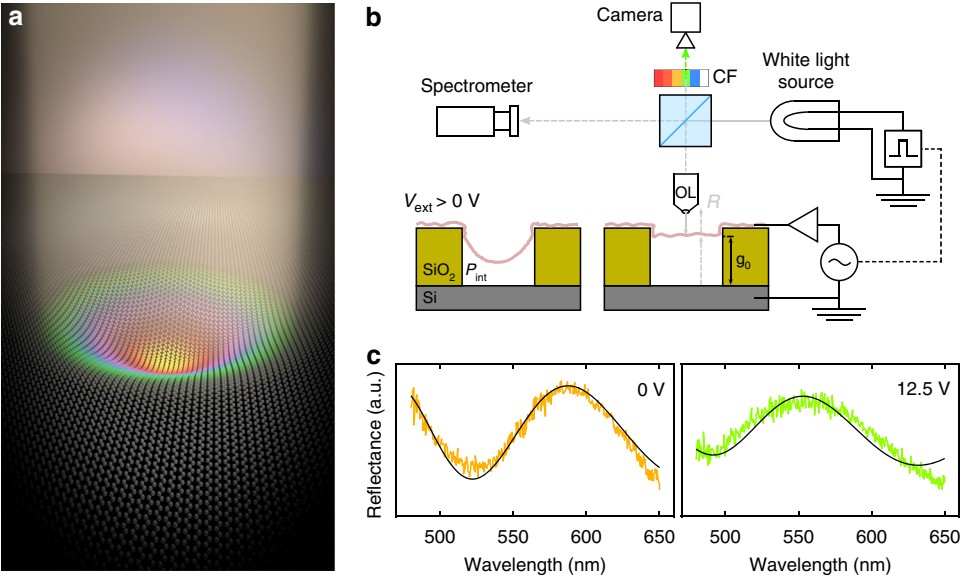

**Fig. 1** Electro-optical measurements with a colorimetry setup of a graphene drumhead of 20 μm in diameter. **a** Artist impression of a deformed CVD DLG drumhead showing Newton's rings upon white light illumination. **b** The membrane can be actuated electrostatically while its reflectance spectrum is measured with a spectrometer and a calibrated camera. The white light source is either a halogen lamp (for spectral studies) or a white LED synchronized to the membrane electrical control (for stroboscopic studies). **c** At different voltages, the reflectance spectrum peak of the overall membrane shifts from 580 nm to 510 nm in a controlled and reproducible manner. Black lines are the theoretical spectral reflectance obtained from Eq. (1)

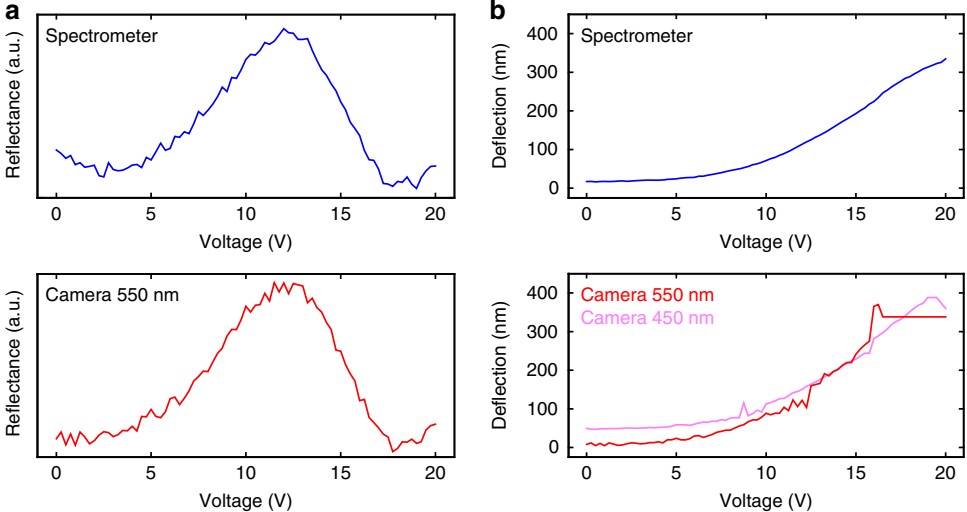

**Fig. 2** Conversion from reflectance–voltage curves to deflection–voltage curves for $\lambda = 550$ nm. **a** The optical response in that selected wavelength from the spectrometer measurements (top panel) and color-filtered camera images (bottom panel) show the same trend. **b** By fitting the reflectance curves with Eq. (1), we obtain the corresponding deflections as a function of applied voltage, which agree well for both types of measurements. A maximum deflection of 350–400 nm is obtained for this membrane (same device as in Fig. 1)

see Supplementary Figure 2). Then, we introduce the optical reflectance[8] by considering the suspended drum as an absorbing layer placed in front of the silicon back mirror while neglecting optical cavity effects. We average the reflectance over the entire drum to obtain the response as perceived by the spectrometer (Supplementary Note 5, and Supplementary Figures 3-4). This drum reflectance can be expressed only in terms of the center deflection as

$$R_{\text{avg}}(\lambda) = A(\lambda) + \frac{B(\lambda)}{\pi \bar{X}}[\sin(\phi) - \sin(\phi - \pi \bar{X})], \quad (1)$$

where $\lambda$ is the wavelength, $A(\lambda)$ and $B(\lambda)$ are wavelength-dependent constants, $\phi = \frac{4\pi g}{\lambda} + \phi'$ is a phase shift induced by the optical travel path and by the graphene, $\hat{X}(r) = \frac{4g\bar{\delta}(r)}{\lambda}$ with $\bar{\delta}$ being the normalized deflection at the center of the drum (i.e., $\bar{\delta} = \delta_{\text{center}}/g$), and $g$ the cavity depth. This model is applied to our devices and shown in Fig. 1c (solid lines).

The electro-optic response of a suspended 20 μm in diameter graphene drum, i.e., the drum averaged reflectance as a function of voltage, is shown in Fig. 2a as obtained from the camera and spectrometer for the 550 nm wavelength. Using Eq. (1), we fit a value of deflection, and plot the deflection vs. voltage in Fig. 2b. The deflection curves from both camera and spectrometer agree well, although the camera fit is done on collected data from a narrow wavelength band (corresponding to that of the filter), while the spectrometer fit is done on a 475–700 nm spectral band. It is worth noting that the fits are insensitive to small deflections and thus to low actuation voltages.

**Stroboscopic measurement.** Given the gas impermeability of graphene[9], these drums should be hermetic, although they could permeate if few defects are present in the membrane[6]. In the previous measurement, a 30 s settling time was used to make sure that no gas permeation effects would interfere with the electro-optic response of the drum, but gas compression effects could alter the position of the membranes when actuating them at sufficiently higher frequencies. Therefore, we investigate the possibility of a squeeze-film like response by fast stroboscopic measurements that are enabled by substituting the halogen lamp

with a white light-emitting diode (LED), which is powered by short duration pulses that are synchronized to a sinusoidal actuation voltage: $V_{\text{act}} = (17.5 + 2.5\sin(\omega_{\text{act}}t + \phi_{\text{act}}))\ V$ (see Methods). By sweeping the phase difference between the illuminating pulse and the actuation signal, we are able to reconstruct the time-domain response of the structure. An example is shown in Fig. 3a where we see two snapshots of a 15 μm drumhead at opposing phases in the actuation cycle.

We stroboscopically measure 5 drumheads of 15 μm in diameter with actuation frequencies ranging from 0.5 to 2000 Hz. The measured structures do not exhibit a first-order low-pass filter response indicative of squeeze-film effect for frequencies up to 400 Hz, beyond which the cut-off frequency of the capacitively loaded amplifier is reached. The frequency response of the drum's center reflectance is shown in Fig. 3b. The absence of a first-order response indicates that the permeation time constants are below or above the measurement range depending on the drumhead in question. A Bode plot of the displacement amplitude and phase are shown in Fig. 3c, demonstrating no special feature until the amplifier cut-off around 400 Hz. Note that this experimental result is limited to 400 Hz due to the cut-off frequency of the electronic amplifier used, rather than an intrinsic limitation of the pixels under atmospheric conditions which we expect to be in the kHz regime.

**GIMOD technology.** Flicker effect in displays is usually perceived negatively and can be reduced or eliminated by increasing the refresh rate[10,11]. The large electro-optic modulation and absence of mechanical delays at frequencies up to 400 Hz make these devices interesting to enable display rates beyond the flicker fusion threshold that require color reproducibility in the millisecond range for applications such as virtual or augmented reality (VR/AR)[12,13]. Figure 4a shows pixels in an ~2500 ppi GIMOD prototype. The static image displaying the Graphene Flagship logo changes color upon the application of a sinusoidal signal (see Supplementary Movie 1). All the mechanical pixels of 5 μm in diameter are addressed at once with the silicon substrate acting as a back gate, hence resulting in areas changing their color except where the graphene is delaminated or broken. The pixel aperture —active to total area ratio—is ~50%, resulting in a reduction of

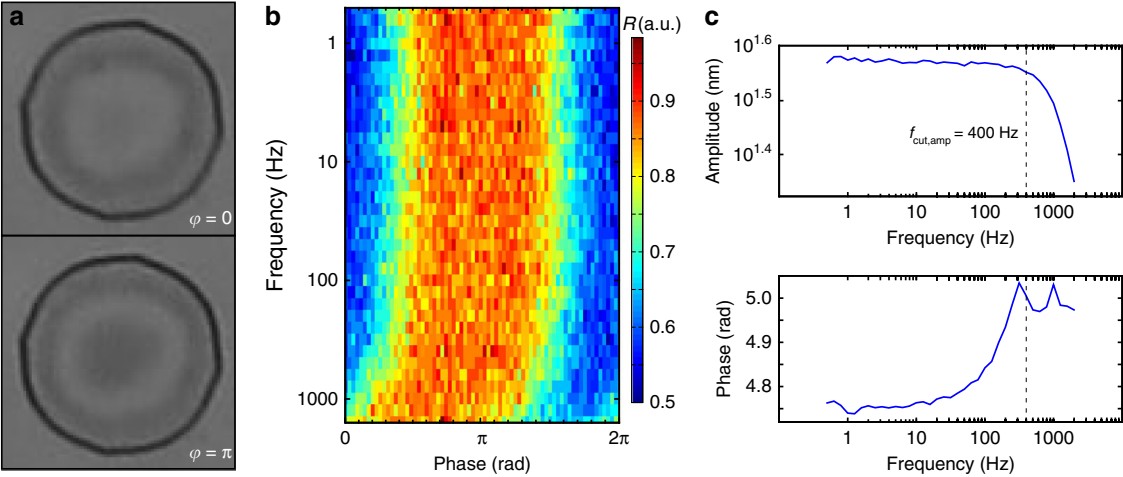

**Fig. 3** Stroboscopic measurement. **a** Optical microscope image of a 15 µm in diameter drum at two opposite phases of vibration ($\phi = 0$ top, $\phi = \pi$ bottom) while actuated at 1 kHz. **b** Color map of the reflectance at the center of the same drum as a function of frequency and phase. A phase delay intrinsic to the signal amplifier is observed. **c** Bode plots of amplitude and phase of the drum, showing that the response is flat up to 400 Hz (cut-off frequency of amplifier)

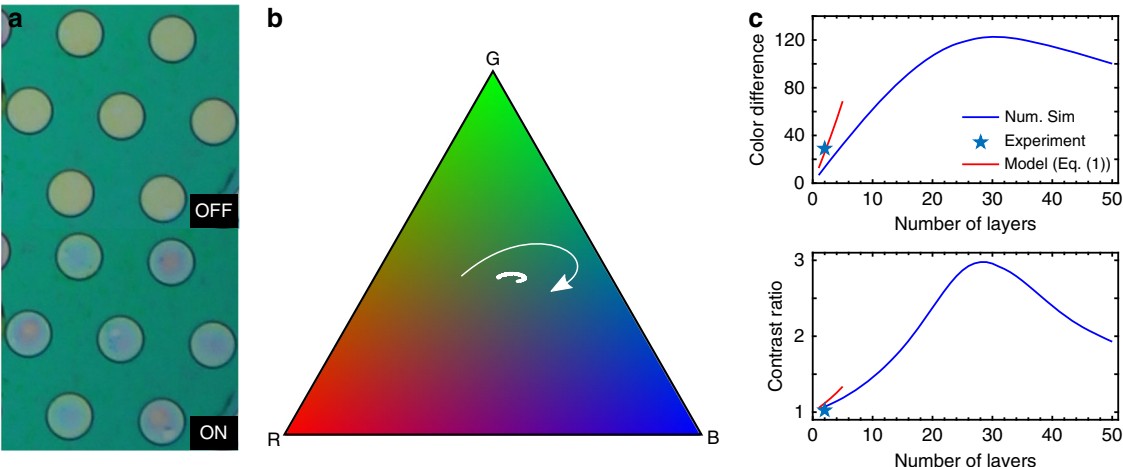

**Fig. 4** Graphene Interferometric Modulator Display (GIMOD) prototype. **a** The graphene mechanical devices are used as continuous-spectrum pixels of 5 µm in diameter, hence resulting in a reflective-type display of 2500 pixels per inch (ppi), equivalent to 12 K+ resolution for a display of 5″. When the GIMOD prototype is OFF, its pixels show a yellow color; when it is ON (30 V), they are in a blue state. **b** Gamut pixel trajectory of the average reflectance of a GIMOD prototype with pixels of 5 µm in diameter. The change of pixel color as a function of voltage, whose direction is pointed with an arrow, is displayed in the sRGB color triangle (subset of x,y chromaticity space) based on CIE1931 colorimetry. **c** Colorimetric response, given by the parameters contrast ratio (CR, bottom panel) and color difference (ΔE, top panel), plotted as a function of the number of graphene layers (for a cavity depth of 400 nm) as obtained from the full model simulations described in SI-8. The plots show an optimum of 29 layers, also shown are the results from the analytical model from Eq. (1) (red) and the response obtained experimentally (star symbol) for comparison

contrast of about 1:2 due to the large separation between pixels (pixel pitch about double of the pixel size). When the pixels are actuated with 30 V they switch from a yellow state (OFF, top panel) to a blue state (ON, bottom panel).

By analyzing the RGB (red, green, and blue) channels of one of the pixels, we obtain its average color gamut and its evolution upon applied voltage. Figure 4b shows the pixel trajectory over a standard RGB (sRGB) color map based on CIE1931 colorimetry[14], demonstrating the electro-optical modulation from yellow to blue of these pixels.

If more graphene layers are added, then the colorimetric response should improve. However, that response is no longer captured by the analytical model in Supplementary Equation 1 (once the number of layers >5), thus simulations based on a more detailed Fabry–Perot model that accounts for optical cavity effects

is undertaken (Supplementary Note 8 and Supplementary Figures 10-11). Based on that model, the colorimetric response of the GIMOD pixels is shown as a function of number graphene layers in Fig. 4c. The model predicts an optimal graphene thickness of 29 layers, equally visible in Fig. 4c is the divergence between the analytical model and the full model with increasing number of layers.

## Discussion

While electrostatic transduction is used extensively as an actuation mechanism for MEMS/nanoelectromechanical system (NEMS) devices, including graphene nanoelectromechanical structures[15–17], in this work we investigate the far-from-resonance, quasi-static, large deformation regime of graphene

membranes[18,19]. For such a regime of operation the hydrostatic pressure component near the zero-deflection position $(\bar{\delta} \sim 0)$ should not be neglected in hermetic cavities as it introduces a significant hydrostatic stiffness (Supplementary Note 4). Only if the membrane is not sufficiently hermetic can this effect be neglected. Moreover, large deflections—even in the case of impermeable membranes—cause the hydrostatic component to be overtaken by the nonlinear stiffness. However, it is possible to have structures with large diameters (>5 μm) and shallow cavities (<300 nm), whereby for a hermetic cavity the hydrostatic stiffness remains dominant over the nonlinear one for all deflections. In such cases both hydrostatic stiffness and gas permeation time can interfere with the proper functioning of the device.

It is interesting to note that among the five 15 μm in diameter membranes measured stroboscopically, one drum shows larger static and dynamic deflections compared to the others (Supplementary Note 6 and Supplementary Figures 5–7). Whether this is due to a lower mechanical stiffness caused by micro-tears, or to a higher membrane permittivity, or to a combination of both (they are not mutually exclusive) remains a question for further investigation.

The measured DC voltage-dependent spectrum is qualitatively similar to simulation results obtained from Eq. (1) (Supplementary Note 5). As it can be also seen in Fig. 2b, the membrane deflection and measured spectrum show little change for low voltages (<5 V). Since drums of 20 μm in diameter are rarely hermetic, this effect is more likely explained by limitations of the technique itself rather than to hydrostatic stiffness. In particular, the assumption of a parabolic profile, although accurate for large deflections, is not necessarily valid for small deformations due to wrinkles[20]. This is a necessary tradeoff in order to fit accurate deflection values for SLG/DLG due to the need to perform a drum average fit rather than a pixel-by-pixel fit as done for thicker membranes[21].

Furthermore, it is interesting to note that according to Eq. (1), the electro-optic response of a GIMOD pixel, when averaged over the drum area, is radius independent. Thus, a change in the radius of the suspended drums does not result in any qualitative modification of the pixel behavior. In addition, the obtained RGB response of the pixels as shown in Fig. 4 shows a limited path across the sRGB triangle. This is confirmed by simulations that predict a maximum change of 10% for each of the RGB channels for a DLG membrane and for an optimal gap for a GIMOD of 600 nm (Supplementary Note 7 and Supplementary Figures 8-9). For a richer colorimetric response, thicker membranes are required, as detailed modeling shows an optimum colorimetric response for 29 graphene layers and a cavity depth around 400 nm.

In summary, we report on CVD double-layer graphene electro-mechanical electro-optic modulators, the characterization of their spectral response, and their application as pixels in a GIMOD. In addition to observing the electro-optical modulation, we develop an electrostatic-optical model to describe their behavior. We use the reflectance–voltage plots to fit a deflection–voltage response and conclude that the device is not significantly hermetic. We further explored the permeation time constants by stroboscopically measuring 5 devices without finding the onset of squeeze-film effect due to the gas trapped in the cavity in the frequency range of 0.5 to 400 Hz. The large color modulation and good frequency response of these graphene electro-optic modulators in ambient conditions enable the use of these mechanical devices as pixels with color reliability at high image refresh rates. This application is demonstrated with a 0.5″ prototype with 2500 ppi that would equate to a 5″ display with 12 K+ resolution.

Besides reducing the motion sickness in VR/AR applications, the demonstrated reflective-type (e-paper) pixel technology would greatly reduce power consumption as it is an IMOD technology. The use of graphene allows the large deformation of small membranes without mechanical failure or hysteresis allowing the fabrication of diffraction-limited devices for ultra-high resolution displays. Moreover, by using different voltages to actuate the pixels, each device is able to generate natural colors in the full spectrum and in a continuous manner. This eliminates the need for RGB subpixels and reduces the addressing bandwidth to only one channel, which enables a further reduction of power consumption.

## Methods

**Colorimetry technique with optical spectrometer**. The deflection as a function of voltage are obtained with the colorimetry technique with an Olympus BX60 microscope as previously reported[6,7]. Pictures are taken with a Canon EOS 600D and a 20× Olympus UMPlanFI objective lens (NA 0.46, LWD 3 mm). A TE-cooled linear CCD array spectrometer (B&W Tek Glacier X) is attached to the optical microscope in such a way that it collects the light reflected from an area slightly larger than the selected drum (about 595 μm$^2$) (Supplementary Note 1 and Supplementary Figure 1). Spectral traces of the entire drum are recorded with an integration time of 20 ms (no averaging). For the electrostatic pulling, we use a power supply (Rigol DP832) to ramp up the voltage from 0 to 20 V with steps of 0.25 V. The voltage is left constant for a period of 30 s after each step and before acquiring the data.

**Stroboscopic technique**. The halogen lamp from the microscope is substituted by a white LED for stroboscopic illumination. A dual channel generator (Keysight Technologies 33512B Waveform Generator) controls the LED by outputting a narrow ($\frac{1}{72}$ of the actuation period) 5 V pulse to the LED. At the same time this illumination pulse is phase-locked to an actuation sine wave that drives the graphene membrane electrostatically.

## Data availability

The authors declare that the data supporting the findings of this study are available within the paper and its Supplementary Information files.

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

## Acknowledgements

The research leading to these results has received funding from the European Union's Horizon 2020 research and innovation programme under grant agreement no. 649953 (Graphene Flagship).

## Author contributions

Devices were fabricated by S.J.C.-B. and D.D. Experiments were designed and performed by S.J.C.-B. and S.H. Materials were designed and synthesized by A.C. and A.Z. Data were analyzed by S.J.C.-B. and S.H. and all authors contributed to the discussion. S.J.C.-B. and S.H. wrote the manuscript, with contributions from all authors.

## Additional information

**Competing interests:** The authors declare no competing interests.

