## [Peer Review File · Nature Communications]

Reviewers' comments:

Reviewer #1 (Remarks to the Author):

The authors of this work have successfully analysed, simulated, and made a prototype of a graphene electrically-actuated pixels for IMODs. This is a really interesting work that could revolutionize the field of display technologies, and the main conclusions and results are solid as demonstrated in the functioning prototype given as a supplementary video. The manuscript is well-written and easy to follow, and the supplementary info contains adequate technical information for reproduction of the work. Here are some of my recommendations on how the manuscript can be improved:

1. One important claim by the authors is that the GIMOD can eliminate perception of flicker due to the high frequency modulation response, but this claim is buried somewhere in page 5. The authors should state this important claim right at the beginning paragraphs and abstract. Also, the authors should expand on this claim with regards to introducing the concept of flicker perception and how GIMOD can suppress such phenomenon.

2. Further on flicker, reference 10 states that flicker perception could only be eliminated with frequencies above 500 Hz, and some colors even reaching 1 kHz. Perhaps the authors should not overextend their claim that GIMOD can eliminate flicker at 400 Hz. A more appropriate claim is the reduction of flicker.

3. In page 6, as stated by the authors and also perceivable from the video, the color contrast is very low due to the large pixel pitch. Can the authors state in their manuscript, the pixel size to pitch ratio required, to get the quality of a standard definition display. Will the improvement in pixel ratio affect the actuation of the graphene pixels?

4. In page 8, stated by the authors and evident in the videos, that the RGB response is limited. It is also stated that thicker membranes can provide a richer colorimetric response. Can the authors provide some further insights on this by:

- i) Give a general descriptive equation on the relationship between graphene membrane thickness and the colorimetric response.
- ii) Give a colorimetric simulation result for thicker graphene membranes
- iii) Recommend the limit of the membrane thickness where graphene may either lose its reflective properties or change in mechanical-actuating behaviour.

Signed,

Dr. Kelvin J. A. Ooi

Reviewer #2 (Remarks to the Author):

The manuscript describes an application of suspended graphene structures. The distance between suspended graphene and the substrate back mirror is modulated electrostatically to demonstrate color display using interferometric absorption. Using a stroboscopic method, the authors also demonstrate that the frequency response of the device is fast enough for most display applications.

I have the following comments/questions:

1. Page 1: "...use DLG membranes because of their higher yiel and their larger absorption in the visible spectrum than that of SLG..." .

Why not use larger number of layers if larger absorption is required? Is there a sweet spot between number of layers that can be modulated and absorption?

2. How does your pixel compare in term of Contrast/gamut to other methods (Ref 5 for instance)?
3. If you select a voltage corresponding to a colour what is the level of monochromaticity that is achieved? It is to some extent answered in figure 4b but some quantitative comparisons would be helpful.
4. For the flagship figure, how is gate voltage applied to only certain portions of the silicon back gate? Is there metallization below the graphene? If I were to take a guess, I would say there is no graphene on the parts that don't change colour in the video. If that is the case, the manuscript should clearly mention it.
5. Clearly you would want the GIMOD to have large viewing angle. What happens if the viewing angle is changed from normal?
6. I find that there is major disadvantage in the scheme proposed compared to Ref 5. Because of the geometry of the device, there is non-uniform sag of graphene. The centre of the drum might move a few 100nm but the rest of the drum doesn't. This makes the colour contrast poor. Some explanation of how this can be overcome and calculations about possible improvement in contrast would be useful.
7. Authors mention "...possibly eliminating the motion sickness in VR/AR applications...". It is not clear how they reached that conclusion? Is it just based on the possible refresh rate of 400Hz? In that case it is a vague statement without any justification.

Reviewer #1

Comment 1: One important claim by the authors is that the GIMOD can eliminate perception of flicker due to the high frequency modulation response, but this claim is buried somewhere in page 5. The authors should state this important claim right at the beginning paragraphs and abstract. Also, the authors should expand on this claim with regards to introducing the concept of flicker perception and how GIMOD can suppress such phenomenon.

We thank the Reviewer for his/her useful comments and discussions. We agree that the reduction of flicker is an important aspect of GIMOD technology, and therefore we have included it in the abstract and extended the discussion on flicker. The introduction and basic characterization of a GIMOD pixel technology is the main novelty of this work, and we would prefer to keep it as the main message of the publication. Frequency modulation issues, including flicker and motion sickness, should be addressed in depth in future publications.

Before: (Page 2) “(...) Based on these findings, we demonstrate a continuous full-spectrum reflective-type pixel technology with a Graphene Interferometric MOdulator Display (GIMOD) prototype of 2500~pixels per inch (ppi) equivalent to more than 12K resolution.”

After: (Page 2) “(...) Based on these findings, we demonstrate a continuous full-spectrum reflective-type pixel technology with **reduced flicker effect by fabricating** a Graphene Interferometric MOdulator Display (GIMOD) prototype of 2500~pixels per inch (ppi) equivalent to more than 12K resolution.”

Before: (Page 5) “The large electro-optic modulation and absence of mechanical delays at frequencies up to 400~Hz make these devices interesting to enable display rates beyond the flicker fusion threshold that require color reproducibility in the millisecond range for applications such as virtual or augmented reality (VR/AR) [10]. (...)”

After: (Page 6) “**Flicker effect in displays is usually perceived negatively and can be reduced or eliminated by increasing the refresh rate [10,11].** The large electro-optic modulation and absence of mechanical delays at frequencies up to 400~Hz make these devices interesting to enable display rates beyond the flicker fusion threshold that require color reproducibility in the millisecond range for applications such as virtual or augmented reality (VR/AR) **[12, 13].** (...)”

- [10] Meehan, Michael, et al. "Effect of latency on presence in stressful virtual environments." *IEEE Computer society*, 2003.
- [11] Kolasinski, Eugenia M. "Simulator Sickness in Virtual Environments. No. ARI-TR-1027". Army research institute for the behavioral and social sciences. Alexandria VA, 1995.
- [12] Hershberger, Wayne A., and J. Scott Jordan. "The phantom array: a perisaccadic illusion of visual direction." *The Psychological Record* 48.1 (1998): 21-32.
- [13] Davis, James, et al. "Humans perceive flicker artifacts at 500 Hz." *Scientific reports* 5 (2015): 7861.

Comment 2: Further on flicker, reference 10 states that flicker perception could only be eliminated with frequencies above 500 Hz, and some colors even reaching 1 kHz. Perhaps the authors should not overextend their claim that GIMOD can eliminate flicker at 400 Hz. A more appropriate claim is the reduction of flicker.

We thank the Reviewer for the kind remark. We accept that “flicker reduction” is more appropriate for the demonstrated performance in this work in view of the new literature included in the manuscript. We would also like to point out that this experimental result is limited to 400 Hz due to the cut-off frequency of the electronic amplifier used, rather than an intrinsic limitation of the pixels under atmospheric conditions which we expect to be in the kHz regime.

Before: (Page 5) “(...) A Bode plot of the displacement amplitude and phase are shown in Figure 3c, demonstrating no special feature until the amplifier cut-off around 400 Hz.”

After: (Page 5) “(...) A Bode plot of the displacement amplitude and phase are shown in Figure 3c, demonstrating no special feature until around 400 Hz. Note that this experimental result is limited to 400 Hz due to the cut-off frequency of the electronic amplifier used, rather than an intrinsic limitation of the pixels under atmospheric conditions which we expect to be in the kHz regime.”

Before: (Page 9) “Besides possibly eliminating the motion sickness in VR/AR applications, the demonstrated reflective-type (e-paper) pixel technology would greatly reduce power consumption as it is an Interferometric MOdulator display (IMOD) technology. (...)”

After: (Page 9) “Besides **reducing** the motion sickness in VR/AR applications, the demonstrated reflective-type (e-paper) pixel technology would greatly reduce power consumption as it is an Interferometric MOdulator display (IMOD) technology. (...)”

Comment 3: In page 6, as stated by the authors and also perceivable from the video, the color contrast is very low due to the large pixel pitch. Can the authors state in their manuscript, the pixel size to pitch ratio required, to get the quality of a standard definition display? Will the improvement in pixel ratio affect the actuation of the graphene pixels?

The pixel size is not standardized in industry as each display and application has different resolution/contrast requirements. We can however state the pixel sizes for, for example, smartphones: 61 μ m in Apple iPhone 7 Plus, 44 μ m in Samsung Galaxy S7, and 31 μ m in Sony Xperia XZ Premium. The size to pitch ratio is typically unreported as customers pay more attention to the final specifications such as resolution. However, we find Ref. 5 talks about a “fill factor” (active area/total area) of 66% while our prototypes have a fill factor of 20%.

Regarding the second question, the authors can think of few reasons why pitch and pixel ratio would affect the actuation or the mechanics of the pixels. These are:

1. *Electrostatics:*

The main issue here would be cross-talk from adjacent pixels. This is not a concern in the current implementation as the entire silicon substrate acts as a background electrode, i.e. the same signal is applied to all pixels simultaneously. In a future implementation of individually addressable pixels, electrostatic crosstalk will need to be modeled and its effect minimized. However, it is possible to say that if the pitch size \gg oxide thickness, the cross-talk will be negligible. Since our simulations indicate

that an optimal gap is on the order of 400 nm, it follows that even for a few μm pitch between pixels, it is possible to ignore electrostatic crosstalk.

2. *Slippage:*

It is interesting to explore if slippage can take place under the effect of electrostatic forcing, whereby the force applied on one pixel would pull on the graphene layer and force it to slip along the silicon oxide surface.

Conceptually, slippage maybe caused by delamination of the graphene from SiO_2 assuming the forces add up in a peeling direction. This can be easily determined to be negligible since literature reports values on the order of 0.6 to 1.3 % strain before delamination []. This is well beyond any realistic value of what can be observed in our pixels (which we estimate from the model in equation S-7 to be well below 0.2% otherwise the pixel would fail) and, even then, we are assuming no crumpling or hidden area [**].*

3. *Finite stiffness of the SiO_2 walls between pixels:*

If the aspect ratio of the SiO_2 isolating the different suspended membranes (i.e. different pixels) becomes extreme, then it may be conceivable that the structure might deform under the effect of lateral electrostatic forces. However, a simple calculation that consider a very extreme aspect ratio of $1 \times 1 \mu\text{m}$ (i.e. pixel separation of $1 \mu\text{m}$ and an oxide thickness of $1 \mu\text{m}$) still results in an extremely stiff structure ($\sim \text{kN/m}$ stiffness for each μm of structure length) whereby we can neglect any deformation due to electrostatics.

- [*] Pérez Garza, H. Hugo, et al. "Controlled, reversible, and nondestructive generation of uniaxial extreme strains ($> 10\%$) in graphene." *Nano Letters* 14.7 (2014): 4107-4113.
- [**] Nicholl, Ryan JT, et al. "Hidden area and mechanical nonlinearities in freestanding graphene." *Physical review letters* 118.26 (2017): 266101.

Comment 4: In page 8, stated by the authors and evident in the videos, that the RGB response is limited. It is also stated that thicker membranes can provide a richer colorimetric response. Can the authors provide some further insights on this by:

- i) Give a general descriptive equation on the relationship between graphene membrane thickness and the colorimetric response.
- ii) Give a colorimetric simulation result for thicker graphene membranes
- iii) Recommend the limit of the membrane thickness where graphene may either lose its reflective properties or change in mechanical-actuating behaviour.

i) Although it is tempting to claim that performance of a GIMOD increases linearly with the number of layers (since more layers absorb more light), the fact is that the colorimetric response needs to be calculated following the procedure detailed in SI. However, it is possible to fit a polynomial relationship to the colorimetric response obtained in the new SI-8 section, and we find that a quadratic relationship best fits the data, as shown below, with the following fit parameters.

From the model:

$$\begin{aligned}\text{Contrast ratio} &= 1.08 + 0.01N \\ \text{Color difference} &= 13.3 + 3N + 1.5N^2\end{aligned}$$

From the full Fabry-Perot simulations:

$$\text{Contrast ratio} = 1.009 + 0.0264N + 0.0018N^2$$

$$\text{Color difference} = 6N - 0.06N^2$$

where N is the number of graphene layers.

ii) We have included a new section in supplementary information (SI-8), where the results obtained from detailed numerical simulations are shown. The new simulations concerning the colorimetric response of GIMOD pixels are done for a variety of graphene thicknesses, and we obtain an optimal graphene layer thickness of 29 layers.

iii) The optical absorption of a N-layer graphene depends on the thickness of the membrane, although a complete optical theory that explains light-2D interactions does not exist currently. It is known that, for few layers ($N < 5$), the absorption in the visible range increases linearly with the number of layers ($A = N \times 2.3\%$) [*], and the absorption of thicker membranes will converge to the bulk value of graphite absorption (around 70%). The interferometric absorption phenomenon in our pixels requires a balance between high absorptions and a sufficiently high transmission for the interference to happen. Our detailed numerical simulations indicate that the optimal performance takes place for 29 graphene layers. Beyond that, the colorimetric response drops quickly indicating the effect of absorption of the thick graphene.

Moving from 2 graphene layers to 29 layers will increase the plate bending rigidity of the structure by a factor of $(29/2)^3 = 3000$. However, if the pixel diameter remains on the order of $10 \mu\text{m}$, then the structure will still act as a membrane rather than a plate [**], i.e. it will still be dominated by tension, thus will not affect the linear behavior significantly. The value of nonlinear stiffness on the other hand increases linearly with the thickness of the graphene membrane, thus an increase of 15 folds in thickness will require a similar increase in the actuation force and thus around 4-fold increase in operating voltage (assuming everything else is equal). Despite this higher operating voltage, which would entail a larger power dissipation during operation, we expect no fundamental change in the behavior of GIMOD pixels.

- [*] Nair, Rahul R., et al. "Fine structure constant defines visual transparency of graphene." *Science* 320.5881 (2008): 1308-1308.
- [**] Cartamil-Bueno, Santiago J., et al. "High-quality-factor tantalum oxide nano-mechanical resonators by laser oxidation of TaSe₂." *Nano Research* 8.9 (2015): 2842-2849.

Reviewer #2

Comment 1: Page 1: "...use DLG membranes because of their higher yield and their larger absorption in the visible spectrum than that of SLG...". Why not use larger number of layers if larger absorption is required? Is there a sweet spot between number of layers that can be modulated and absorption?

We thank the Reviewer for the useful questions and comments. Our current fabricated devices were obtained by stacking multiple layers of graphene, after they have been grown separately, on top of each other over the cavities. This is both a delicate and a time-consuming process, increasing the number of layers also increases the chances that some failure will happen in this artisanal way of producing devices. Although a systematic study regarding the experimental characterization of the impact of the number of layers on the colorimetric response is extremely interesting, it is nevertheless to be done once more reliable and streamlined manufacturing procedures are developed.

To address the Reviewer's point we added a new section where numerical simulations are performed to estimate the optical response of GIMOD pixels as a function of number of layers.

Comment 2: How does your pixel compare in term of Contrast/gamut to other methods (Ref 5 for instance)?

Although it would be interesting to compare both technologies, Ref 5 does not indicate how the CR was calculated. But assuming they use the same definition as we do, i.e. $\max L^*/\min L^*$, we obtain an optimal contrast ratio of 3 compared to 30 for the MEMS-based technology. Note that if the contrast ratio is measured with respect to a uniform illumination spectrum (i.e. averaged equally over all wavelength without accounting to the peculiarities of human vision), then we obtain a contrast ratio of 6.

Comment 3: If you select a voltage corresponding to a colour what is the level of monochromaticity that is achieved? It is to some extent answered in figure 4b but some quantitative comparisons would be helpful.

The authors are not clear on what the reviewer means by monochromaticity. If that term refers to color purity as defined by the RGB color standard, then the answer would not only depend on the optical response of the GIMOD but also on the spectral nature of the illuminating source. This is why, when choosing a source for the numerical analysis the authors chose a Halogen-Tungsten lamp, since that is the source used experimentally. Different illumination conditions can give different color perception experiences. The trajectory on the RGB triangle in Figure 4 of the manuscript is based on the RGB components of the measured GIMOD pixels. These components are plotted below on a Cartesian coordinate for convenience.

If on the other hand monochromaticity implies spectral purity, as in the spectral reflectance as a function of voltage, then this is given by our simplified model, equation 1 in the main text, and plotted in Figure 1 for both model and experimental

data. As the number of graphene layers increases, the spectral response of the pixel will then be described by the more complete model in section 8 of SI.

Comment 4: For the flagship figure, how is gate voltage applied to only certain portions of the silicon back gate? Is there metallization below the graphene? If I were to take a guess, I would say there is no graphene on the parts that don't change colour in the video. If that is the case, the manuscript should clearly mention it.

The Reviewer is right in that the areas of the display prototype that do not change the color are areas where the graphene delaminated or broke. The gate voltage is applied through the silicon bottom layer of the display and we cannot control different areas of the displays in these prototypes. We have added a sentence in the caption of Figure 4 to avoid misunderstandings:

Before: (Page 6) “(...) All the mechanical pixels of 5 μm in diameter are addressed at once with the silicon substrate acting as a back gate. (...)”

After: (Page 6) “(...) All the mechanical pixels of 5 μm in diameter are addressed at once with the silicon substrate acting as a back gate, hence resulting in areas changing their color except where the graphene is delaminated or broken. (...)”

Comment 5: Clearly you would want the GIMOD to have large viewing angle. What happens if the viewing angle is changed from normal?

The reviewer raises an interesting point in asking about the effect of viewing angle. The obvious effect of viewing angle would be to increase the distance light travels, and thus changes slightly the interference pattern, and thus the perceived color. For small angles, the additive path length can amount to an increase of up to 10% in the effective cavity depth. The effect of this can be seen on the contrast and colorimetric response of the GIMOD pixel in our new SI-8 section, Figure S-11. The effect is non-negligible, but this holds true for all interferometric display technologies regardless of the material and pixel geometry.

Comment 6: I find that there is major disadvantage in the scheme proposed compared to Ref 5. Because of the geometry of the device, there is non-uniform sag of graphene. The centre of the drum might move a few 100nm but the rest of the drum doesn't. This makes the colour contrast poor. Some explanation of how this can be overcome and calculations about possible improvement in contrast would be useful.

We thank the Reviewer for his/her comment. The reviewer is correct in pointing out the inherent limitation of using a circular clamped membrane geometry which results in a displacement profile that ends up reducing the contrast and possible RGB color compared to the above mentioned reference where a square geometry with flexural linkages is used.

However it is important to point out that unlike the very mature MEMS fabrication process, constructing such complicated mechanical structures out of graphene membranes is beyond what an academic institution (and possibly industry) can achieve at the moment. The required post-transfer lithography and etching, and the

failure rate of such structures are for now unknown quantities. Even though a square freestanding geometry is probably not an ideal solution for GIMOD, we agree that a solution needs to be found that would enhance the performance of these pixels. While we consider other geometries and more complex structures to be beyond the reach of this paper, we have investigated the simple solution of increasing the number of layers in a new SI section 8.

Comment 7: Authors mention “..possibly eliminating the motion sickness in VR/AR applications...”. It is not clear how they reached that conclusion? Is it just based on the possible refresh rate of 400Hz? In that case it is a vague statement without any justification.

We thank the Reviewer and apologize for the confusion. It is known that flicker artifacts can induce motion sickness [] and should not exist at the frequencies we experienced [**]. However, flicker effect is one of the many factors causing motion sickness. We have made several corrections to make this message clear by claiming a “flicker reduction” rather than “elimination”, which is more appropriate for the demonstrated performance in this work in view of the new literature included in the manuscript.*

- [*] Kolasinski, Eugenia M. *Simulator Sickness in Virtual Environments*. No. ARI-TR-1027. Army research inst for the behavioral and social sciences Alexandria VA, 1995.
- [**] Hershberger, Wayne A., and J. Scott Jordan. "The phantom array: a perisaccadic illusion of visual direction." *The Psychological Record* 48.1 (1998): 21-32.

Before: (Page 5) “(...) A Bode plot of the displacement amplitude and phase are shown in Figure 3c, demonstrating no special feature until the amplifier cut-off around 400~Hz.”

After: (Page 5) “(...) A Bode plot of the displacement amplitude and phase are shown in Figure 3c, demonstrating no special feature until around 400 Hz. Note that this experimental result is limited to 400 Hz due to the cut-off frequency of the electronic amplifier used, rather than an intrinsic limitation of the pixels under atmospheric conditions which we expect to be in the kHz regime.”

Before: (Page 9) “Besides possibly eliminating the motion sickness in VR/AR applications, the demonstrated reflective-type (e-paper) pixel technology would greatly reduce power consumption as it is an Interferometric MODulator display (IMOD) technology. (...)”

After: (Page 9) “Besides reducing the motion sickness in VR/AR applications, the demonstrated reflective-type (e-paper) pixel technology would greatly reduce power consumption as it is an Interferometric MODulator display (IMOD) technology. (...)”

List of main changes made to the manuscript:

- Major modification of Figure 4
- Major modifications in main text about Figure 4

Before: (Caption Figure 4) “Graphene Interferometric MOdulator Display (GIMOD) prototype showing the logo of the Graphene Flagship. The graphene mechanical devices are used as continuous-spectrum pixels of 5 μ m in diameter, hence resulting in a reflective-type display of 2500 pixels per inch (ppi), equivalent to 12K+ resolution for a display of 5". When the GIMOD prototype is OFF, its pixels show a yellow color; when it is ON (30 V), they are in a blue state. b) Gamut pixel trajectory of the average reflectance of a GIMOD prototype with pixels of 5 μ m in diameter. The change of pixel color as a function of voltage, whose direction is pointed with an arrow, is displayed in the sRGB color triangle (subset of x,y chromaticity space) based on CIE1931 colorimetry.”

After: (Caption Figure 4) “Graphene Interferometric MOdulator Display (GIMOD) prototype. **a)** The graphene mechanical devices are used as continuous-spectrum pixels of 5 μ m in diameter, hence resulting in a reflective-type display of 2500 pixels per inch (ppi), equivalent to 12K+ resolution for a display of 5". When the GIMOD prototype is OFF, its pixels show a yellow color; when it is ON (30 V), they are in a blue state. b) Gamut pixel trajectory of the average reflectance of a GIMOD prototype with pixels of 5 μ m in diameter. The change of pixel color as a function of voltage, whose direction is pointed with an arrow, is displayed in the sRGB color triangle (subset of x,y chromaticity space) based on CIE1931 colorimetry. **c) Colorimetric response, given by the parameter contrast ratio (CR, bottom panel) and color difference (ΔE , top panel), plotted as a function of the number of graphene layers (for a cavity depth of 400 nm) as obtained from the full model simulations described in SI-8. The plots show an optimum of 29 layers, also shown are the results from the analytical model from equation s1 (red) and the response obtained experimentally (star symbol) for comparison.”**

Before: (Page 6) “Figure 4a shows the substrate of a ~2500 ppi Graphene Interferometric MOdulator Display (GIMOD) prototype. The static image displaying the Graphene Flagship logo changes color upon the application of a sinusoidal signal (see Supplementary Video S10). All the mechanical pixels of 5 μ m in diameter are addressed at once with the silicon substrate acting as a back gate. The pixel aperture - active to total area ratio- is ~50%, resulting in a reduction of contrast of about 1:2 due to the large separation between pixels (pixel pitch about double of the pixel size). The panels in the middle show a zoom-in of some pixels when they are actuated with 30 V (yellow state when OFF, top panel; blue state when ON, bottom panel).

By analyzing the RGB channels of one of the pixels, we obtain its average color gamut and its evolution upon applied voltage. Figure4b shows the pixel trajectory over a standard RGB (sRGB) color map based on CIE1931 colorimetry [11], demonstrating the electro-optical modulation from yellow to blue of these pixels. (...)”

After: (Page 6) “Figure 4a shows **pixels in** a ~2500 ppi Graphene Interferometric MOdulator Display (GIMOD) prototype. The static image displaying the Graphene Flagship logo changes color upon the application of a sinusoidal signal (see Supplementary Video **S12**). All the mechanical pixels of 5 μ m in diameter are addressed at once with the silicon substrate acting as a back gate. The pixel aperture -

active to total area ratio- is ~50%, resulting in a reduction of contrast of about 1:2 due to the large separation between pixels (pixel pitch about double of the pixel size). When the pixels are actuated with 30 V they switch from a yellow state (OFF, top panel) to a blue state (ON, bottom panel).

By analyzing the RGB channels of one of the pixels, we obtain its average color gamut and its evolution upon applied voltage. Figure 4b shows the pixel trajectory over a standard RGB (sRGB) color map based on CIE1931 colorimetry [11], demonstrating the electro-optical modulation from yellow to blue of these pixels.

If more graphene layers are added, then the colorimetric response should improve. However, that response is no longer captured by the analytical model in equation S1 (once the number of layers > 5), thus simulations based on a more detailed Fabry-Perot model that accounts for optical cavity effects is undertaken (Supplementary Information 8). Based on that model the colorimetric response of the GIMOD pixels is shown as a function of number graphene layers in Figure 4c. The model predicts an optimal graphene thickness of 29 layers, equally visible in Figure 4c is the divergence between the analytical model and the complete model is also visible with increasing number of layers. (...)"

Before: (Page 8) "(...) This is confirmed by simulations that predict a maximum change of 10% for each of the RGB channels for a DLG membrane and for an optimal gap for a GIMOD of 600 nm (Supplementary Information Section 7). For a richer colorimetric response, thicker membranes may be required. Ultimately graphene mechanical pixels need to be optimized taking into consideration the yield of the fabrication process, the actuation voltage, and the colorimetric response."

After: (Page 8) "(...) This is confirmed by simulations that predict a maximum change of 10% for each of the RGB channels for a DLG membrane and for an optimal gap for a GIMOD of 600 nm (Supplementary Information Section 7). For a richer colorimetric response, thicker membranes are required, as detailed modeling shows an optimum colorimetric response for 29 graphene layers and a cavity depth around 400 nm."

- Minor modifications of sentences in the main text.
- New references.
- New section 8 in Supplementary Information.

REVIEWERS' COMMENTS:

Reviewer #1 (Remarks to the Author):

I'm very satisfied with the authors' responses to my questions. Especially for my 4th comment, that the authors have performed quite a convincing numerical analysis for the optimal number of layers of graphene for the maximum contrast and color response. I would strongly recommend the publication of this paper in its current form, and I wish the authors the best of luck to develop the 29-layer GIMOD prototype in their subsequent studies.

Signed, Dr. Kelvin J. A. Ooi